# The Evolution in Non-Alcoholic Fatty Liver Disease Patients’ Profile and the Associated Sustainable Challenges: A Multidisciplinary Perspective

**DOI:** 10.3390/nu16111584

**Published:** 2024-05-23

**Authors:** Maridi Aerts, Zenzi Rosseel, Elisabeth De Waele

**Affiliations:** 1Department of Gastroenterology and Hepatology, Universitair Ziekenhuis Brussel (UZB), 1090 Brussels, Belgium; maridi.aerts@uzbrussel.be; 2Department of Pharmacy, Universitair Ziekenhuis Brussel (UZB), 1090 Brussels, Belgium; 3Department of Clinical Nutrition, Universitair Ziekenhuis Brussel (UZB), 1090 Brussels, Belgium; elisabeth.dewaele@uzbrussel.be; 4Faculty of Medicine and Pharmacy, Vrije Universiteit Brussel (VUB), 1090 Brussels, Belgium

**Keywords:** cirrhosis, nutrition, sustainability, waste, NAFLD

## Abstract

The prevalence and incidence of NAFLD is rising due to the obesity pandemic, caused by the widespread availability of ultra-processed foods and the decrease of physical activity. Factors such as socioeconomic status (SES), ethnicity and geographical location are associated with NAFLD, with lower SES correlating with higher incidence, particularly in regions like America or Europe. Beside the quality of food, the quantity also plays a crucial role. The World Health Organization (WHO) recommends a Mediterranean diet with a balanced energy intake. Since no hard medical treatment is available for NAFLD, lifestyle adjustments are key. Patient empowerment by providing relevant information and co-ownership of the therapy will increase the implementation rate and enhance the quality of medical follow-up and medication adherence, as studies report a good adherence to medication among patients who are well-aware of the severity of their disease. Regarding sustainability, patients with NAFLD have a high load of ambulatory follow-up, which, since the COVID-19 pandemic, can be partially provided by teleconsulting. Both patients’ lifestyle modifications and healthcare practitioners’ therapeutical strategy can decrease the carbon footprint.

## 1. Introduction

Non-alcohol fatty liver disease (NAFLD) represents a spectrum of liver diseases characterized by fat accumulation unrelated to alcohol consumption. The global prevalence of NAFLD is increasing rapidly and is a major cause of chronic liver disease [1]. Obesity, sedentary lifestyle, smoking and ultra-processed foods are some of the contributing factors in the development of NAFLD [2]. Recent data highlighted those geographical disparities with regions exhibiting a high incidence of metabolic syndrome, like North America, increase the prevalence of NAFLD. Understanding these geographical differences is crucial to tailoring public health interventions [3] and decrease the substantial burden on health economics. The easy access to ultra-processed foods with a high amount of saturated lipids and excessive sugars are contributing to rising numbers of NAFLD [4,5,6]. Such unhealthy dietary food choices are leading to disruptions of lipid metabolism, insulin resistance and inflammation [7]. A healthy diet and regular exercise play a pivotal role in counteracting NAFLD [8]. Excessive food production, consumption and waste are contributing to environmental unsustainability. It is therefore important to address food waste and to create sustainable practices that involve changes in dietary patterns and reduce the environmental impact [9,10]. Healthcare professionals play a crucial role in steering patients towards medical adherence, but patients themselves can actively contribute to their treatment. Empowering patients with knowledge, responsibility and correct information is integral to the comprehensive management of NAFLD [11,12].

## 2. Methods

A bibliographic search in PubMed, Scopus and Web of Science was conducted for this narrative review, with publications published between 2000 and 2024 included. This search was based on a combination of MeSH and free terms: ‘cirrhosis’, ‘NAFLD’, ‘diabetes’, ‘metabolic syndrome’, ‘obesity’, ‘socioeconomic’, ‘smoking’, ‘sleep behavior’, ‘exercise’, ‘nutrition’, ‘food waste’, ‘sustainability’, ‘lifestyle’, ‘patient empowerment’ and ‘interventions’. Additional papers were identified by snowballing. Articles were discussed and referred to if they were published in English between 2000 and 2024 and if they contained information suitable for discussion regarding the topic of this article. Exclusion criteria were studies in infants and neonates, on liver diseases other than NAFLD and liver cirrhosis, articles not published in English or Dutch and published before 2000. In order to find articles that investigated the relationship and therefore the influence of nutrition and lifestyle adaptations on the incidence of NAFLD and liver cirrhosis, Boolean operators (“AND”/“OR”) were used. The data extraction method contained scientific publications assessed to be eligible for this review by the authors that were organized and structured by topic. Additional papers were withheld by citation chaining.

## 3. Chronic Liver Disease

NAFLD affects an estimated 38% (95% CI 33.71–42.49) of adults worldwide [1,13] and around 13% of children and adolescents [14]. This is linked to chronic liver disease, with its prevalence rising alongside obesity rates [15]. When left unmanaged, the disease will progress through various stages, such as steatosis, steatohepatitis, fibrosis/cirrhosis and hepatocellular carcinoma. It is closely associated with metabolic syndrome and independently contributes to cardiovascular disease (CVD) risk [16]. Since 2023, there is a global discussion about the disease nomenclature. There have been several suggestions, aiming to shift away from the term ‘non-alcoholic’ and instead provide a name that more accurately represents the metabolic origins of the disease. The term MASLD (Metabolic dysfunction Associated Steatotic Liver Disease) highlights the connection between metabolic dysfunction and the development of fatty liver disease, in particular the role of underlying metabolic factors in the pathogenesis.

Managing NAFLD requires addressing both liver and extra-hepatic manifestations of this metabolic syndrome such as diabetes. When we look at the relevance of environmental pollution, there is increasing evidence that various environmental contaminants, such as persistent organic pollutants (POPs), endocrine-disrupting chemicals (EDCs), heavy metals, and micro- and nano–plastics, contribute to the onset and advancement of NAFLD. Apart from dietary exposure, inhaling pollutants poses another potential risk for NAFLD, particularly in heavily urbanized areas, where these exposures may compound with other risk factors, such as an obesogenic food environment, disruption of circadian rhythms and decreased physical activity levels [17].

The climate crisis is one of the biggest threats to global health, and the interaction between climate change and NAFLD is underestimated [18]. Not only obesity and its metabolic consequences but also malnutrition can lead to the development of fatty liver [19,20]. In the absence of pharmaceutical treatment options, other modalities need to be explored. The focus shifts to enhancing insulin sensitivity and promoting weight loss, where necessary. A holistic approach is indicated, including behavioural change counting habit building techniques and patient empowerment. Relevant domains are nutrition, socioeconomic status, exercise, smoking and sleep quality.

Additionally, efforts concentrate on reducing the inflammatory environment associated with obesity, as these factors primarily underlie the progression of the disease [21,22]. Our goal is to provide insights on how we can make sustainable choices to improve the outcome of these NAFLD patients. Sustainability is about making choices that are not only good for you but also consider the broader impact on the environment and community. Small, consistent changes in your lifestyle can contribute to both personal health and the well-being of the patient.

## 4. Lifestyle

### 4.1. Nutrition and Profile

The European Association for the study of the liver (EASL) and the European Society for Clinical Nutrition and Metabolism (ESPEN) play a crucial role in ensuring adequate nutritional therapy in patients with liver diseases, particularly those with cirrhosis. In addition to the EASL and ESPEN guidelines, various other sets of guidelines exist [19,20,23,24]. A systematic review evaluated 13 clinical practice guidelines (CPG) [25]. The review concluded that the EASL and ESPEN guidelines had the highest score for scope and purpose. They featured a multidisciplinary stakeholder panel, and their key suggestions were easily distinguishable throughout the text, highlighting the quality and effectiveness in guiding nutritional interventions [25].

Several large trials and the ESPEN guidelines already demonstrated the negative outcomes in patients suffering from malnutrition [20,26,27,28,29]. Malnutrition prevalence in cirrhosis patients ranges from 5% up to 92%. Lack of knowledge about malnutrition or difficulties in nutritional assessment and diagnosis are contributing to this wide range. A trial including 187 cirrhotic patients evaluated the nutritional intake and food habits as well as the one- and two-year mortality. Seventy-three per cent had a Child-Pugh class A and only 4.6% were severely malnourished, according to subjective global assessment rating (SGA). The study cohort had an energy and protein intake of 20.1 kcal/kg/day and 0.88 g/kg/day. Compared to the EASL and ESPEN guidelines, which recommend 35 and 30–35 kcal/kg/day and 1.2–1.5 g/kg/day of proteins [30]. The EASL does not recommend the use of SGA for nutritional assessment, and underestimates the prevalence of muscle loss, and the agreement is rather low compared to other methods (total lymphocyte count, BMI and handgrip measurement) [19,24].

Patients with NAFLD have a typical profile analysed by two systematic reviews. The first review included 92 studies and concluded that patients with NAFLD have a mean age of 48 years (38–59) and a BMI of 25.8 kg/m^2^ (22.3–30.4) [1]. A second review, including 34 studies, revealed the highest incidence of NAFLD in Hispanic people, intermediate in white people and lowest in Black people [31]. There are significant geographical variations in the rates of NAFLD prevalence, as well as the severity of NAFLD and non-alcoholic steatohepatitis (NASH), and associated complications. These differences can be attributed to a combination of genetic factors and sociodemographic determinants. The advancement of socioeconomic status (SES) in regions characterized by low or middle sociodemographic indexes (SDI) is closely associated with increased availability and consumption of unhealthy diets. Adverse outcomes related to NAFLD are often linked to the excessive intake of sugar- and fructose-rich foods in Latin America, independent of rising rates of obesity and type 2 diabetes. Conversely, in certain parts of the world such as Southern Europe, the adoption of a Mediterranean diet has been recognized as an intervention to prevent or manage NAFLD. Gaining a precise understanding of the evolving epidemiology of NAFLD on a global scale presents challenges due to the lack of access to high-quality data, particularly in regions such as most African countries. Epidemiological investigations into NAFLD are frequently from a North American perspective, which complicates the global picture [32].

The importance of nutrition extends beyond the qualitative aspect, as quantity also plays a crucial role. Recent numbers stated by the World Health Organization (WHO) of 2023 revealed that 1.9 billion adults are overweight or obese while 462 million are underweight. Underweight as well as obesity are categorized under malnutrition and are posing a challenge in the 21st century. There is a rise in healthcare expenses due to malnutrition related diseases such as obesity, cardiovascular diseases and hypertension [33]. The most important contributing factor of malnutrition is an imbalance of energy. The energy imbalance between calories consumed and calories expended is nominated as one of the primary reasons for the obesity pandemic stated by the WHO. Recent data showed that obesity is often associated with NAFLD [34]. To reduce those high numbers of malnutrition and the incidence of NAFLD, the WHO made some recommendations about sugar and fat intake. It is important that total fat does not exceed 30% of total energy intake, and saturated fats and free sugars should be limited to 10% of total energy intake [33]. Notably, your SES, religious beliefs and geographical location will determine the type and quantity of food you will consume. To help with nutritional choices, the WHO recommendations are there for guidance. Nutritional changes and a healthy diet are key factors in the treatment of NAFLD.

### 4.2. Socioeconomic

SES and lifestyle are independently correlated with each other. The lower the SES, the harder it is to have access to healthy foods, plan regular mealtimes and exercise on a regular basis [35]. A recent trial with 5272 patients revealed that patients with a high SES consume less alcohol, had a lower incidence of diabetes and hypertension and exercised regularly compared to patients with a middle or low SES. Not only health-related factors were significantly different also the hepatic steatosis index and comprehensive NAFLD score were lower [36]. SES is not the sole factor that has an influence on lifestyle—also ethnicity and geographical location have an impact. The frequency of different nutritional patterns (vegetables, fish, meat, rice, soy) may vary across the world [37]. Ultra-processed foods are characterized by a high number of sugars and saturated fatty acids, leading to a higher risk of developing obesity, cholesterol and metabolic syndrome [38]. Apart from geographical location and ethnicity, religion might also impact the consumption of pork, alcohol and meat [37]. What you eat has an impact on muscle strength, the immune system and cognitive function [39,40,41]. It is therefore important to follow a diet that includes a variety of vegetables, fruits, alternate fish and meat, low in saturated FA and with a minimum of ultra-processed foods [42].

### 4.3. Exercise

Lifestyle changes include more than changes in nutritional pattern alone. Incorporation of regular exercise, smoking cessation and adequate sleep can reduce the incidence and worsening of NAFLD.

Patients often have a combination of factors contributing to the development of NAFLD as stated in Figure 1. One of those factors is obesity, already addressed by the WHO as the ‘obesity pandemic’. To tackle this factor, it is important to incorporate exercise into daily life. This exercise can be aerobic or anaerobic, with or without weight loss or in the form of resistance training. With aerobic exercise (running, swimming) being focused on cardiovascular conditioning, and resistance training (weightlifting) focused on muscle strength improvement, people can combine training patterns [21,43,44,45,46]. The EASL recommend 150–200 min of exercise per week but did not specify what kind of exercise [47]. A systematic review analysing the influence of exercise on liver parameters and cholesterol in NAFLD concluded that there was a reduction in aminotransferases, low-density lipoprotein cholesterol (LDL-C) and triglycerides (TG) with overall exercise, while aerobic exercise caused a reduction in aminotransferases and intrahepatic lipids. On the other hand, resistance training had a positive effect on triglycerides and total cholesterol [47]. A second systematic review analysed the influence of exercise on the evolution of liver parameters, but on long term (>4 weeks). The results were in accordance with Babu et al. [47]: a decrease in aminotransferases, LDL-C and TG was observed [45]. Both studies accounted for 23 studies compiling 1322 patients [45,47] and showed that incorporation of exercise in daily living has a proven benefit on liver parameters and therefore on improving outcomes in patients with NAFLD.

### 4.4. Smoking

Recent trials investigated the link between smoking and the presence of NAFLD. Cigarette smoke contains more than 7000 chemicals, with a huge proportion known to be harmful and related to cardiovascular diseases, diabetes type 2 and NAFLD [48,49,50]. Especially the activation of sterol regulatory element-binding protein, leading to synthesis of fatty acids, is associated with NAFLD. A large Korean trial, including 9603 patients, examined the association between a history of smoking and NAFLD. This study revealed an association between the presence of NALFD and being an ex-smoker and between active smokers with a short smoking cessation (<10 years) or >10 pack years [49]. Besides this large trial, a systematic review was conducted including 28 studies, resulting in a patient population of 4,465,862 patients, to reveal a possible correlation between the presence of NAFLD in former and current smokers; former smokers had an increase in BMI, because smoking and BMI have an inverse relationship. The BMI increase after smoking cessation can be explained by the fact that smoking increases energy expenditure and suppresses caloric intake [50]. The increase in BMI and the evolution to obesity is one of the factors contributing to the development of NAFLD. Former smokers had no lower chance to develop NAFLD in contrast to current smokers. This study concluded that smoking cessation was not a protective factor against NAFLD [48].

### 4.5. Sleep

Apart from exercise and smoke cessation, a good nights’ rest is of cardinal importance. The occurrence of repetitive sleep deprivation periods has a negative impact on health. An interrupted night rest leading to nocturnal snacking has a negative effect on the blood sugar levels and increases the low-density lipoprotein to high-density lipoprotein cholesterol ratio [51]. A U-shaped curvilinear relationship was observed between sleep duration and BMI in women, but in men there was a monotonic trend towards higher BMI with shorted sleep duration demonstrated in a trial with 1040 participants [52,53]. Also, levels of ghrelin increased in patients with shorter sleep duration. Ghrelin is a hormone that regulates appetite causing nocturnal snacking and thus waves in insulin levels [52]. A large Japanese study with 2172 patients investigated the relationship between sleep duration and NAFLD. Sleep apnoea was higher in men and women with NAFLD, leading to sleep disturbances, compared to patients without NAFLD [53]. Furthermore, shorter sleep duration was positively associated with the incidence of NAFLD. This association was also addressed in a large meta-analysis and prospective trial with thousands of patients [54,55].

### 4.6. Patient Empowerment

As no hard medical interventions are available for NAFLD, lifestyle adaptations are key components in both prevention and cure. This is where the active role of the patient is of uppermost importance (Figure 2). Patient empowerment (PE) has received a lot of popularity in the 21th century, with 64,265 hits on PubMed^®^ since it was advocated by the WHO in 2012. The WHO stated that patient-centred care should be a key element for improving healthcare outcomes, facilitating a better communication between patients and healthcare practitioners (HCP) and better use of primary health resources [56]. In the case of cirrhotic patients, disease complexity, the high number of drugs, lack of education and misconceptions are commonly reported barriers to treatment [57]. Five trials, all including cirrhotic patients, assessed baseline knowledge about their condition. They all made interventions, from an information leaflet or video to in-person educational sessions to improve disease understanding. They concluded that baseline knowledge was low and was improved by the interventions [58,59,60,61,62]. Patient health empowerment and quality of live (QoL) are correlated to each other, as proved by a study with 30 cirrhotic patients. QoL and health-promoting lifestyle profile scores were evaluated after health education using health empowerment theory and compared to a control group. The study group had a significantly higher awareness of major clinical symptoms, diet and use of medication. This improved the activities of daily living and correlated to the QoL [63]. To facilitate PE, there is the help of technology. There is a wide range of apps accessible for patients to take their therapy in own hands and to communicate with HCP from a distance. Doctors and other HCPs can answer questions and can guide patients throughout their drug treatment.

## 5. Science Implementation

(Nutritional) Science implementation is the translation of research into clinical practice, from bench to bedside. The objective of this translation is to develop an adequate nutrition plan to optimize nutritional adequacy. This process must tackle barriers, weaknesses and be applicable in real-world conditions [64]. Translating guidelines and medical treatments into clinical practice is not always as easy as it seems. Cirrhotic patients need a multidisciplinary approach. This involves pharmaceutical guidance to ensure medication adherence, an adapted nutritional policy and a consistent follow-up by the gastroenterologist [63,65]. The HCP needs to follow specific guidelines designed for this patient group and communication between HCPs is of cardinal importance. To reduce in-hospital mortality, increase quality of care and decrease medication errors, standardized order sets were implemented [66,67,68]. Order sets are an integration of evidence-based practices and standardized protocols, and have proven to be instrumental in achieving an effective healthcare system. Those standardized order sets have an impact on the initiation and continuation of therapy. Successful therapeutic interventions are dependent on accurate prescriptions and proper dosing of medication with sets obviating the need for handwritten prescriptions and the risks associated with manual documentation. Complex therapies will be handled with greater precision and consistency, enhancing patient safety [66]. This innovative approach will lead to a higher healthcare standard for patients and HCPs. Implementation of care bundles, like order sets, specific for NAFLD patients were documented in the UK. The care bundle was compiled based on recommendations of EASL and National Institute for Health and Care Excellence (NICE). It provides a checklist in order to record anthropometrical data, metabolic risk factors and weight reduction targets. Fifty patients entered the trial but in only 46% of the patients this bundle was used. The bundle was associated with a significantly better documentation and implementation of aspects of patient management [69].

Last, there is a need for quality indicators (QI) that measure the quality of treatment and knowledge of cirrhosis to measure impact on feedback in a sustainable way [70]. Those QIs cover different domains, from aetiology, variceal bleeding, general health care, vaccination state to screening of hepatocellular carcinoma [70,71,72]. An American trial, which reviewed 265 medical charts from cirrhotic patients, concluded that sending reminders (scans, blood samples, vaccination) every six months to medical doctors increased compliance and with that quality of therapy [70]. An Australian study, including 302 cirrhotic patients, selected eight ascite-specific QIs. Adherence to these QIs was high and ranged between 70% and 92%. Compliance to lower 30-day readmission rates were due to early paracentesis and early initiation of diuretic therapy [72].

## 6. Interventions and Their Success Rates

### 6.1. Lifestyle

Guidelines for managing NAFLD generally include recommendations related to lifestyle modifications, dietary interventions, and potential pharmacological treatments. One of the primary recommendations when managing NAFLD is weight management through a combination of dietary changes and increased physical activity. Even a small weight loss (around 5–10% of body weight) can have positive effects on liver health. Not only weight loss, but also increasing insulin resistance and increasing postprandial metabolism, are interesting endpoints. One study measured the effect of a low-fat vegan diet and showed, beside a reduction in body weight, also a reduction in hepatocellular and intramyocellular fat and an increased insulin sensitivity [73]. Patients are advised to start some physical activity, including both aerobic activities and resistance training. This contributes not only to weight loss but also to overall metabolic health. Since these lifestyle changes may ask a lot of effort of these patients, they may need ongoing support and follow-up to maintain these lifestyle changes, so regular monitoring of liver function and other relevant parameters is essential. In some cases, especially when lifestyle modifications are not sufficient to lose weight, pharmacological interventions may be considered. However, these decisions are usually made on a case-by-case basis, and medications may have associated risks and benefits. Currently, there are no regulatory approved pharmacological treatments for NAFLD, making lifestyle interventions, particularly calorie restriction for weight loss and disease regression, crucial in NAFLD management. While dietary changes are effective for reducing CVD and Type 2 Diabetes risk, achieving recommended weight loss targets is challenging for many patients. Cirrhotic patients face challenges such as fluid and salt restrictions and suffer from an impaired bile acid metabolism [20,74] leading to hepatic fat accumulation. This is linked to insulin resistance and steatosis, which are typical side effects of NAFLD and worst-case scenario progression to cirrhosis [19,24]. The Mediterranean diet (MD) is promising in preventing and managing NAFLD [20,42], but the optimal dietary pattern and achievable adherence remain unclear [75]. The green-Mediterranean diet is a diet restricted in red/processed meat, enriched with green plants and polyphenols. One study showed that this green-Med diet can double intrahepatic fat loss in comparison with other healthy strategies, and could reduce NAFLD by half [76]. A cross-sectional trial with 101 cirrhotic patients examined the association between adherence to a Mediterranean diet and the severity of NAFLD. The adherence score was significantly higher in non-smokers and patients with diabetes and hypertension. The study revealed that a higher intake of fruits, in particular, was associated with a lower severity of NAFLD, as indicated by the fibrosis score [20,77]. Incorporation of environmentally friendly and long-term practices into the patients’ lifestyle could be beneficial in making the management of NAFLD more sustainable.

### 6.2. Pharmacological

A healthy lifestyle including MD and regular exercise are key components in controlling NAFLD and potential evolvement to cirrhosis. Correlated to the limitation of cirrhotic evolvement is medication adherence. Several trials evaluated the medication adherence, the possible complications of non-adherence and the reasons for it. Medication adherence decreases as the therapy progresses over time, as stated by a trial with hepatitis B cirrhotic patients [78]. Less than half of patients reported that they never miss a medication moment. Reasons for non-compliance reported were forgetfulness (42%), being away from home (36%) and running out of medication (25%). There was no correlation between adherence and knowledge of disease, pill burden and beliefs about effectiveness [79]. This was also stated by Hayward et al., who interviewed 100 patients with cirrhosis and evaluated therapy adherence, perception of illness and QoL. Medication adherence was high in 42%, moderate in 37% and low in 21%. In the group with low adherence, patients had a lower perception of treatment helpfulness or could not afford their medication. A lower QoL with symptoms such as abdominal pain and shortness of breath was reported in that same group [80]. High compliance rates were reported and knowledge of the disease was not correlated to non-compliance. It is the role of healthcare practitioner (HCP) to make the patient aware of the importance of regular appointments, examinations and the correct intake of their medication. Increasing not only compliance but also raising awareness about the severity of the disease is of primordial importance. Patients should therefore always be properly informed about what the disease entails, what the consequences are and in correlation with that disease, and the possible negative outcomes. Failure to follow recommendations and guidelines lead to deterioration of clinical status, resulting in a higher mortality rate [19].

## 7. Sustainability and Food Waste

The environmental impact of hospital visits for NAFLD patients is significant and should not be overlooked. It is crucial to minimize these visits to only those that are absolutely necessary, while ensuring patient safety and avoiding further hospitalizations. Finding alternative means of patient contact that do not require hospital visits is imperative. The COVID-19 pandemic has facilitated progress in this regard, as telemedicine and video consultations have become integrated into our practice [81], offering effective solutions for remote patient care. Patients with cirrhosis experience significant unmet psychosocial, practical and physical needs and tend to have a diminished quality of life and increased utilization of healthcare services, leading to substantial costs and a high carbon footprint [19]. Interventions aimed at fulfilling these unmet needs could potentially mitigate hospital healthcare expenses and carbon footprint. Future research should investigate whether addressing patients’ needs can serve as an efficient approach to encouraging a more cost-effective utilization of healthcare services, potentially resulting in reduced hospital admissions and emergency presentations.

Besides the core business of caring and curing patients, a hospital environment can contribute to sustainability. Hospitals can help mitigate greenhouse gas emissions by reducing food waste. The production, transportation and disposal of food require significant resources, including water, energy and land. Minimizing this food waste can contribute to overall resource conservation. Hospitals adopting sustainable practices, such as efficient food purchasing, meal planning and waste reduction strategies, can play a role in reducing their environmental footprint. There is limited information about food waste in hospitals, but we do know that there are many points that can be addressed. Monitoring and improving hospital services could improve food intake and reduce food waste by applying changes in the service system, menus, serving time, patient needs, training staff, communications, quality of food, and meal conditions. This can lead to increased compliance with meals and a reduction in food waste [82]. Another study, by Berardy et al., showed that the mean total food waste was significantly higher with meat-containing-meal consumers than vegetarian-meal consumers [83]. Since hospitals are increasingly recognizing the importance of sustainability, addressing food waste aligns with broader efforts to promote environmental stewardship and community well-being.

## 8. Conclusions

Nutritional interventions are lifesaving in cirrhotic patients. Challenges exist and change due to the changing nature of the patients. Keeping in mind global challenges including the green deal, a sustainable approach is of cardinal importance and needs to be incorporated in both clinical practice and research initiatives.

## Figures and Tables

**Figure 1 nutrients-16-01584-f001:**
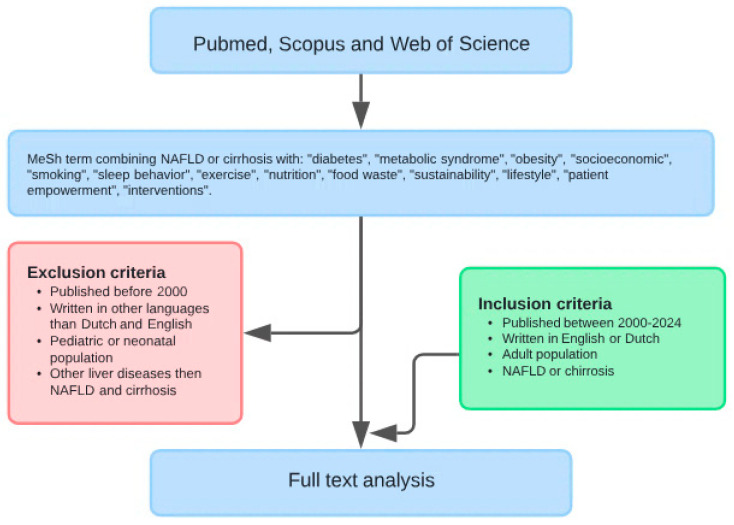
This flowchart illustrates the procedure for the selection of relevant articles according to predefined in- and exclusion criteria. After the initial screening, full texts were analysed, summarized and compared to existing literature.

**Figure 2 nutrients-16-01584-f002:**
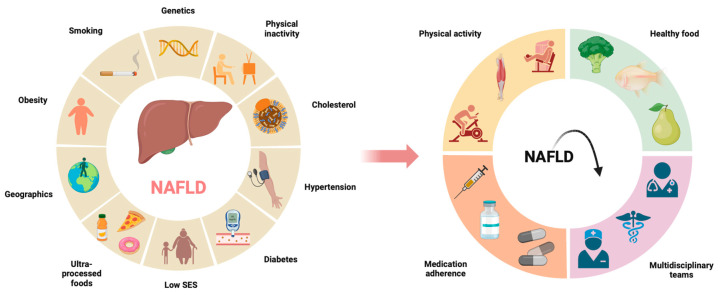
Predisposing factors in the development of NAFLD and possible solutions to slow down and prevent the occurrence of NAFLD. The circle diagram on the left describes possible factors that can contribute to the development of NAFLD. The development of NAFLD results often from a combination of these factors. The circle diagram on the right refers to possible solutions in preventing and slowing down progression of NAFLD.

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
