# Peer review of "The Evolution in Non-Alcoholic Fatty Liver Disease Patients’ Profile and the Associated Sustainable Challenges: A Multidisciplinary Perspective"

_nutrients, 2024, doi:10.3390/nu16111584_

Round 1

Reviewer 1 Report

Comments and Suggestions for Authors

The manuscript was well written.

There are some minor comments.

1. Regarding the methodology, it would be beneficial to incorporate more search terms related to NAFLD, like metabolic syndrome and diabetes. It's also advisable to broaden the scope of the study to include various fields such as nutrition, psychology, and sociology.

2. The manuscript would be enhanced by a detailed and clear explanation of the search strategy, selection criteria, and data extraction method.

3. Please explain Figure 1 in detail.

Comments on the Quality of English Language

Please check English grammar.

Author Response

Dear reviewer

Thank you for the valuable information and comment. 

Please find the answers to your comments in the attachment. 

Regards

The authors

Reviewer 2 Report

Comments and Suggestions for Authors

Dear Authors:

Regarding the manuscript with title “The evolution in non-alcoholic fatty liver disease patients' profile and the associated sustainable challenges: a multidisciplinary perspective”, I have two major concerns.

Comment 1:

Authors wanted to do a multidisciplinar perspective regarding non-alcoholic fatty liver disease (NAFLD). As authors correcty presented on Figure 1, there are several predisposing factors of NAFLD. Almost all manuscript is focused only in the relationship between Nutrition and NAFLD.

The subchapter 5 refers to Lifestyle. Lifestyle includes includes physical activity, diet, smoking and sleep behaviors. In this subchapter, authors only talk about diet.

The subchapter 7 refers to Interventions and their success rates. One more time, authors only present diet interventions.

I presente several interventions regarding the other predisposing lifestyle factors that of NAFLD

Exercise:

Liu, Y., Xie, W., Li, J., & Ossowski, Z. (2023). Effects of aerobic exercise on metabolic indicators and physical performance in adult NAFLD patients: A systematic review and network meta-analysis. Medicine102(14), e33147. https://doi.org/10.1097/MD.0000000000033147

Gao, Y., Lu, J., Liu, X., Liu, J., Ma, Q., Shi, Y., & Su, H. (2021). Effect of Long-Term Exercise on Liver Lipid Metabolism in Chinese Patients With NAFLD: A Systematic Review and Meta-Analysis. Frontiers in physiology12, 748517. https://doi.org/10.3389/fphys.2021.748517

Babu, A. F., Csader, S., Lok, J., Gómez-Gallego, C., Hanhineva, K., El-Nezami, H., & Schwab, U. (2021). Positive Effects of Exercise Intervention without Weight Loss and Dietary Changes in NAFLD-Related Clinical Parameters: A Systematic Review and Meta-Analysis. Nutrients13(9), 3135. https://doi.org/10.3390/nu13093135

Smoking:

Jang, Y. S., Joo, H. J., Park, Y. S., Park, E. C., & Jang, S. I. (2023). Association between smoking cessation and non-alcoholic fatty liver disease using NAFLD liver fat score. Frontiers in public health11, 1015919. https://doi.org/10.3389/fpubh.2023.1015919

Zhang, S., Liu, Z., Yang, Q., Hu, Z., Zhou, W., Ji, G., & Dang, Y. (2023). Impact of smoking cessation on non-alcoholic fatty liver disease prevalence: a systematic review and meta-analysis. BMJ open13(12), e074216. https://doi.org/10.1136/bmjopen-2023-074216

Kim, D., Vazquez-Montesino, L. M., Li, A. A., Cholankeril, G., & Ahmed, A. (2020). Inadequate Physical Activity and Sedentary Behavior Are Independent Predictors of Nonalcoholic Fatty Liver Disease. Hepatology (Baltimore, Md.)72(5), 1556–1568. https://doi.org/10.1002/hep.31158

Sleep behaviors:

Mukherji, A., Dachraoui, M., & Baumert, T. F. (2020). Perturbation of the circadian clock and pathogenesis of NAFLD. Metabolism: clinical and experimental111S, 154337. https://doi.org/10.1016/j.metabol.2020.154337

Yu, L., Lin, C., Chen, X., Teng, Y., Zhou, S., & Liang, Y. (2022). A Meta-Analysis of Sleep Disorders and Nonalcoholic Fatty Liver Disease: Potential Causality and Symptom Management. Gastroenterology nursing : the official journal of the Society of Gastroenterology Nurses and Associates45(5), 354–363. https://doi.org/10.1097/SGA.0000000000000658

Comment 2:

On Methods, authors must state how many manuscripts were found by authors and how many were used at the end (inclusion and exclusion criteria). I also suggest authors to add other terms to find manuscripts related to physical activity, sleep behaviors and smoking. MESH terms could be used to extend the search.

Comments on the Quality of English Language

Minor editing of English language required.

Author Response

(The authors gave the same response as above.)

Round 2

Reviewer 2 Report

Comments and Suggestions for Authors

Dear Authors.

Regarding the manuscript with title “The evolution in non-alcoholic fatty liver disease patients' profile and the associated sustainable challenges: a multidisciplinary perspective”, despite the changes made by authors, I still have the same concerns that, in my opinion, hindered the publication of this Review.

Regarding my previous Comment 1, authors only add information regarding the other lifestyle factors (exercise, sleep bahaviors and smoking) on subchapter 5. The remaing chapters were written only according to the relationship between nutrition aspects and non-alcoholic fatty liver disease. Authors have to reorganize the Review according to several lifestyle factors.

Regarding my previous Comment 2, authors must specify how the search strategy was done.  The operators AND or OR must be added to the search strategy to allow the replication of the search. Although not mandatory in a Review, it will be important that authors present a flowchart to a better comprehension of the sequence and steps since the identification of studies to the inclusion of studies in the review. Authors only add to the revised manuscript a very broad information ragerding the inclusion criteria in which they stated “if they contained information suitable for discussion regarding the topic of this article.”. No exclusion criteria was presented.

Author Response

Dear reviewer

Please find the responses in the attachment. 

regards

the authors 
